# Compression Therapy Is Not Contraindicated in Diabetic Patients with Venous or Mixed Leg Ulcer

**DOI:** 10.3390/jcm9113709

**Published:** 2020-11-19

**Authors:** Giovanni Mosti, Attilio Cavezzi, Luca Bastiani, Hugo Partsch

**Affiliations:** 1Angiology Department, MD Barbantini Clinic, Via del Calcio n.2, 55100 Lucca, Italy; 2Eurocenter Venalinfa, 63074 San Benedetto del Tronto (AP), Italy; info@cavezzi.it; 3Institute of Clinical Physiology, Italian National Research Council e CNR, 56100 Pisa, Italy; lucabastiani@ifc.cnr.it; 4Dermatologic Department, Medical University of Vienna, 1010 Vienna, Austria; hugo.partsch@meduniwien.ac.at

**Keywords:** venous leg ulcers, mixed leg ulcers, diabetes mellitus, compression therapy

## Abstract

The aim of this study was to investigate if compression therapy (CT) can be safely applied in diabetic patients with Venous Leg Ulcers (VLU), even when a moderate arterial impairment (defined by an Ankle-Brachial Pressure Index 0.5–0.8) occurs as in mixed leg ulcers (MLU). Materials and methods: in one of our previous publications we compared the outcomes of two groups of patients with recalcitrant leg ulcers. Seventy-one patients were affected by mixed venous and arterial impairment and 109 by isolated venous disease. Both groups were treated by tailored inelastic CT (with compression pressure <40 mm Hg in patients with MLU and >60 mm Hg in patients with VLU) and ultrasound guided foam sclerotherapy (UGFS) of the superficial incompetent veins with the reflux directed to the ulcer bed. In the present sub analysis of the same patients we compared the healing time of 107 non-diabetic patients (NDP), 69 with VLU and 38 with MLU) with the healing time of 73 diabetic patients (DP), 40 with VLU and 33 with MLU. Results: Twenty-five patients were lost at follow up. The results refer to 155 patients who completed the treatment protocol. In the VLU group median healing time was 25 weeks for NDP and 28 weeks in DP (*p* = 0.09). In the MLU group median healing time was 27 weeks for NDP and 29 weeks for DP (*p* = −0.19). Conclusions: when providing leg ulcer treatment by means of tailored compression regimen and foam sclerotherapy for superficial venous refluxes, diabetes has only a minor or no effect on the healing time of recalcitrant VLU or MLU.

## 1. Introduction

Compression therapy (CT) is a therapeutic mainstay in leg ulcer treatment. Compression therapy, whatever compression device is used, affects all deformable structures of the area where it is applied. In the leg, it will compress arteries, veins, lymphatics and tissue so exerting several effects [1]. Compression of veins will produce venous narrowing [2] reducing venous pooling and increasing blood flow velocity. This will produce a reduction in venous reflux [3] and an increase in venous pumping function [4]. At microcirculatory level compression of venous and and lymphatic capillaries will reduce capillary filtration and improve lymphatic drainage. By these effects, compression therapy is able to effectively treat leg edema and lyphedema [5,6,7,8]. In addition, a positive effect on inflammatory cytokines was demonstrated as compression is able to decrease the inflammatory mediators and increase their antagonists [9,10]. Finally, it has been shown that compression is able to increase the arterial flow both in normal volunteers and in patients with arterial impairment provided the compression pressure will not exceed the arterial pressure [11].

Due to all the described effects, compression therapy is extremely effective in the treatment not only of VLU but also of MLU and of vasculitic ulcers. The only true contraindication is represented by critical limb ischemia. Compression therapy in diabetic patients is a debated topic. Many authors consider diabetes mellitus as an exclusion criterion in enrollment of patients requiring CT [12]. On the other hand, the lack of papers reporting data about CT in DP with leg ulcers seems to demonstrate a widespread skepticism on application of this therapeutic procedure. As a result, we were able to find just one paper on CT in DP with leg ulcer [13]. In this study a healing rate of 81% and of 67% was described in leg ulcers without and with additional arterial impairment, respectively. In one of our previous papers [14] we reported a retrospective observational analysis on the treatment of 180 outpatients with recalcitrant ulcers (so defined by the absence of any sign of healing after 6 months treatment). In this paper we retrospectively compared the healing time of 109 patients affected by VLU with the healing time of 71 patients with MLU treated in an outpatient setting between January 2011 and July 2014. Patients with mixed ulcers were characterized by a moderate peripheral arterial occlusive disease (PAOD) with an ankle brachial pressure index (ABPI) between 0.5 and 0.8.

The aim of the present study was to perform a sub analysis of the available data from that study, to verify if compression therapy has any impact in terms of effectiveness and safety in diabetic patients with VLU and MLU.

## 2. Patients and Methods

### 2.1. Patients

In this sub-analysis we assessed the healing time of the same 180 patients (43 males and 137 females; mean age was 74 ± 11.5 years; age ranged between 31 and 92 years) of the previous study taking into account the co-existence of diabetes mellitus type II as a potential confounding variable.

Inclusion criteria: patients of both sex and any ages affected by VLU due to superficial or deep venous reflux, without and with moderate PAOD, in inflammatory stage defined as ulcer bed partially or totally covered by necrotic slough with or without clinical signs of local infection; ulcer size up to 100 cm [2]; ulcer duration more than 6 months; no signs of healing tendency. Patients with no insulin dependent diabetes mellitus (nIDDM) were included in the study

Exclusion criteria: small ulcers, ulcer surface >100 cm [2]; ulcer duration shorter than 6 months; insulin dependent diabetes mellitus (IDDM); patients on immunosuppressive drugs therapy; active cancer; life expectancy lower than 6 months; severe PAOD (ABPI < 0.5).

Both in the VLU and MLU groups the healing time of NDP was compared with that of DP.

Patients were examined by color duplex ultrasound (My Lab 60 with a multi-frequency linear probe (7.5–12 MHz); Esaote s.p.a., Genoa, Italy) in standing position to investigate the superficial and deep veins and in supine position to investigate the arterial system of the lower limbs. Reflux in deep veins, saphenous veins, and tributaries was elicited in the standing patient by manual calf compression/release and by the Valsalva maneuver. Reflux time >0.5 s for superficial veins and >1 s for the deep veins was considered pathological. Venous occlusion/obstruction was diagnosed by ultrasound compression. Systolic ankle and brachial blood pressure were measured in every patient, using an 8 MHz continuous-wave Doppler probe. Ankle Pressure Brachial Index (ABPI) was calculated by dividing the systolic ankle/brachial pressure.

All subjects gave their informed consent for the proposed treatment. An ethical committee approval is not requested in Italy for retrospective observational studies involving patients who were submitted to routine clinical treatment.

### 2.2. Topical Treatment

All the patients were treated in an outpatient setting once per week on average. All ulcers were treated with the same cleansing saline solution and with the same polyurethane foam dressing. Cadexomer powder (Smith & Nephew, Hull, UK) was added in patients with locally infected ulcers (defined as blocked healing, redness of peri-wound skin, increasing fluid exudate and/or pain, change in granulation tissue appearance, or bad smelling) until the clinical signs of infection disappeared. No patient received systemic antibiotic treatment.

### 2.3. Compression Therapy

All patients were treated by CT using inelastic materials applying the bandages from the base of the toes to the knee in a spiral fashion. In the patients with VLU the compression device consisted of a short stretch bandage (Rosidal K; Lohmann & Rauscher, Rengsdorf, Germany) applied with full stretch on top of a sub-bandage padding layer made up of cotton padding and a multi-layer cohesive short stretch bandage (Cellona and Mollelast Haft (both Lohmann & Rauscher)).

In patients with MLU, Cellona and Mollelast Haft were applied with reduced stretch (“modified compression”), and Rosidal K was not used. Bandages were exclusively applied by very well trained and experienced staff, in most cases by the doctor (GM).

### 2.4. Compression Pressure

Compression pressure (CP) was measured by using a Picopress device (Microlab Elettronica sas, Padua, Italy). CP was measured in the first 4 weeks of treatment, both after application and before removal of the bandage. The probe was applied at the B1 point (the transition from gastrocnemius tendon to gastrocnemius muscle on the medial aspect of the leg) as recommended by a consensus document [15]. CP was set to the range of about 60 mm Hg at application in patients with VLU both in NDP and in DP and of about 40 mm Hg in patients with MLU both in NDP and in DP. Bandage removal and dressing changes were planned once a week. Patients were asked to return for additional visits in the event of unusual pain, excess exudate, or any unwanted effect.

### 2.5. Vein Ultrasound Guided Foam Sclerotherapy

All the patients with superficial venous reflux directed to the ulcer bed were submitted to ultrasound guided foam sclerotherapy (UGFS) [16,17] with 3% sodium tetradecyl sulfate in the saphenous trunk and 1% in the tributaries. Sclerosant foam was prepared according to the Tessari method and a liquid to gas ratio of 1:4.

### 2.6. Ulcer Pain

Ulcer pain was assessed by using a Visual Analogue Scale (VAS) at every visit.

### 2.7. Statistical Analysis

Values were expressed as medians, with interquartile range (IQR) and minimum and maximum values. Non-parametric tests were used to compare the two groups. The Mann–Whitney test was used to compare the baseline characteristics of the VLU and MLU patients and to compare CP. The healing time was compared by Kaplan–Meier analysis and the log-rank test. Finally, we performed two multivariate Cox proportional hazard models for the VLU and MLU patient groups to study the healing time (dependent variable) in relation to diabetic and not-diabetic patients (independent variables). The analysis was adjusted for sex, age, smoking habit, arterial hypertension, body mass index and ulcer surface and the results was shown as unadjusted hazard ratios (HRs).

*p* value < 0.05 was considered as threshold for statistical significance. IBM Statistical Package for Social Sciences (SPSS, version 22, Chicago 2013) and Prism 6 (GraphPad, CA, USA) were used for statistical analysis and to generate graphs.

## 3. Results

In the VLU group the healing time of 69 NDP (13 males, 56 females, age 72.8 ± 12.6 years) was compared with that of 40 DP (12 males, 28 females, mean age 70.5 ± 15.7 years). In the MLU group the healing time of 38 NDP (9 males, 29 females, age 76.9 ± 9 years) was compared with that of 33 DP (13 males, 20 females, age 74 ± 11.3 years). (Figure 1).

Sixteen patients in the VLU group (11 NDP and 5 DP) were lost at follow-up. In the MLU group four NDP were lost at follow up, three patients underwent amputation (one NDP and two DP) and two DP died (Figure 1). Results refer to 155 patients (93 in the VLU group (58 NDP and 35 DP) and 62 in the MLU group (33 NDP and 29 DP). NDP and DP in both the groups of VLU and MLU showed comparable demographic characteristics (Table 1).

### 3.1. Healing Time

In the VLU group the median healing time was 25 weeks for NDP and 28 weeks for DP (*p* = 0.09) In the MLU group median healing time was 27 weeks for NDP and 29 weeks for DP (*p* = 0.19). Fifty-two weeks was the maximal healing time in one diabetic patient with MLU (Figure 2). According to Kaplan–Meier analysis DP had a delayed healing time compared with NDP in both VLU and MLU groups (Figure 3), but the difference between the two groups was not statistically significant (*p* = 0.06).

The Multivariate Cox model analysis was performed separately for the VLU and MLU groups (adjusted for sex, age, smoking habit, hypertension, BMI and ulcer surface) (Table 2). In the VLU group, NDP showed a shorter healing time compared with DP (HR 0.460 (95% CI, 0.284–0.746) *p* = 0.002). No differences were found for sex, age, smoking habit, BMI > 30 or arterial hypertension. The smaller the ulcer size the shorter the healing time (HR 0.946 (95% CI, 0.933–0.959) *p* < 0.000). In the MLU group, the healing time of NDP was shorter compared to DP but the difference is not statistically significant. (HR 0.760 (95% CI, 0.418–1.382) *p* = 0.368). Again, patients with a smaller ulcer surface healed in a shorter time (HR 0.897 (95% CI, 0.873–0.923) *p* < 0.000). Females showed a longer healing times than males (HR 0.457 (95% CI, 0.225–0.930) *p* = 0.031), while no differences were found for age, smoking habit, BMI > 30 or arterial hypertension.

### 3.2. Compression Pressure

In VLU patients (both in NDP and in DP) median CP at bandage application was 65 mm Hg in supine position and 85 mm Hg in standing position. In MLU patients (again both in NDP and DP) CP at bandage application was 40 mm Hg in supine position and 58 mm Hg in standing position with a statistically significant difference between the two pressure ranges (*p* < 0.001). Both compression devices showed a significant pressure drop at bandage removal; median values of 35/51 mm Hg in VLU and 25/44 mm Hg in MLU were recorded in the lying/standing position respectively, 7 days after the original application of the multilayer compression system (Figure 4).

### 3.3. Foam Sclerotherapy

An average of 8 ± 2 mL (maximum dose 12 mL) of sclerosant foam per session was injected under ultrasound guidance. In VLU patients 50 NDP (86%) and 28 (70%) DP were submitted to UGFS of their superficial vein reflux as well as 32 NDP (97%) and 18 DP (55%) in the MLU group. An average of 1.8 sessions (range 1–3) was necessary to achieve the complete occlusion of the target venous segment.

### 3.4. Symptoms

Diabetic patients were found to be negative in simple tests for sensory disturbance using needle pricks or a tuning fork; in addition, they hardly reported typical symptoms of neuropathy (numbness, reduced ability to feel pain, tingling or burning, sharp pains or cramps, hyperesthesia, sweating not temperature-related).

In both groups, patients did not report any compression-related pain at bandage application. Some tightness sensation due to the strong pressure was reported by patients with VLU at bandage application. This symptom decreased overtime and came close to zero at removal time. Both DP and NDP with MLU tolerated the applied modified compression very well and did not complain about any compression-related discomfort. Ulcer-related pain was significantly greater (*p* < 0.0001) in patients with MLU than in patients with VLU, but no difference was found in subgroups of NDP and DP. Pain decreased overtime as reflected by VAS that gradually decreased to zero after 8–12 weeks in patients with MLU, and in 4 weeks in patients with VLU; in fact, no difference was found in reported pain between NDP and DP after 12 weeks.

We performed the Posteriori Power Analysis for Venous Leg (VLU) and mixed leg ulcers (MLU) groups. The Posteriori Power Analysis was based on the difference of means of healing time between NDP and DP. For the VLU group, with 58 NDP and 35 DP, the estimated power for a two-sample means test was above 0.60 (0.62). For MLU group, with 33 NDP and 29 DP, the estimated power for a two-sample means test was above 0.50 (0.57).

## 4. Discussion

Due to the increasingly higher incidence of diabetes in recent years, its prevalence in patients affected by chronic venous diseases is growing. As a consequence, the co-existence of diabetes in VLU and MLU raises questions in terms of treatment, especially concerning compression therapy that has been demonstrated to be effective in favoring VLU and MLU healing. Compression therapy has long been considered risky practice in patients with diabetes because of the fear of compromising the arterial system. Indeed, diabetes and related disorders, such as metabolic syndrome, are directly linked with degeneration of arterial wall and the development of a general cardio–cerebro–vascular risk [18]. Diabetic wounds, impacted by insufficient angiogenesis, show decreased vascularity and capillary density [19]. Wound closure is frequently delayed in diabetes and chronic non-healing wounds are common. Angiogenesis impairment is associated with a low level or downregulation of vascular endothelial growth factor (VEGF). VEGF has been shown to be one of the most important angiogenic factors in wounds and its level is reduced in diabetic wounds [20,21]. In addition, ulcers in diabetic patients show a complete derangement of inflammation mediators: interleukin (IL)-1β, IL-6, tumor necrosis factor-α (TNF-α). IL-10, an anti-inflammatory mediator, and TGF-β, exerting positive effects on wound healing, are downregulated [22,23]. Besides all these deficiencies at the microcirculatory level, diabetes has a negative impact also on the arterial macrocirculation increasing the arterial stiffness so that a normal ABPI does not exclude an impairment of the arterial nutritional flow in diabetics [24].

Because of all the factors mentioned above and due to the clinical experience of delayed wound healing, diabetes was considered an a priori exclusion criterion for compression therapy in some ulcer healing studies [12]. However, these facts do obviously play a minor role when a correct clinical and ultrasound-based diagnostic process is put in place, and the deciding pathophysiology adequately treated including compression therapy. Diabetic patients with venous leg ulcers, both without and with moderate impairment of large arteries, treated by compression therapy and foam sclerotherapy of venous reflux, showed an overall slightly delayed healing compared with non-diabetic patients; this difference was not statistically significant and no specific safety issues were encountered in this study.

For a diabetes-specific wound-healing problem, the improvement of angiogenesis and vascular perfusion probably represents the core goal. For a long time, CT was regarded as a contraindication in all clinical situations when an arterial impairment occurs. It is a common belief that compression pressure would impair the flow of nutritional vessels and decrease peripheral perfusion. Some recent literature questioning these dogmas show an increase, rather than a decrease, in the arterial perfusion under adequate compression [11,25,26]. Even in diabetes-specific clinical conditions, like diabetic neuropathic foot ulcers, may benefit from compression by using intermittent pneumatic compression (IPC) devices, as a form of “aggressive edema reduction” [27].

One common mechanism of action of compression in these indications is the reduction in edema, by which the distances between nutritional capillaries and the tissue cells will diminish. In patients with leg edema this may be achieved by low pressure [6] that can be exerted even by special low-pressure stockings that have been proposed in diabetic patients [28].

In patients with mixed, arterial-venous ulcers with an ABPI > 0.5 and ankle perfusion pressure >60 mm Hg, it was demonstrated an increase in the arterial inflow as long as the exerted pressure (by properly applied inelastic bandages) did not exceed 40 mm Hg [11,26]. This indicates that PAOD represents a real contraindication for CT only in case of critical limb ischemia with an ankle pressure < 60mm Hg, irrespective of diabetes occurrence. In case of a media sclerosis that is quite frequent in diabetics, other methods to assess the arterial inflow, e.g., toe pressure measurement, should be employed; a toe pressure value above 30 mm Hg indicates that “modified compression” not exceeding 40 mm Hg may be safe [11].

In addition to its effects on arterial and venous system, CT is effective in decreasing inflammatory mediators, such as interleukin (IL)-1β, IL-6, tumor necrosis factor-α (TNF-α) and in increasing anti-inflammatory mediators, such as IL-10 and TGF-β so counteracting the negative effect of diabetes on these mediators [9,10].

Finally, IPC was shown to upregulate VEGF, thereby contributing to improve neovascularization that is connected to ulcer healing [29,30]. Nevertheless, inelastic bandages exert a higher pressure by moving from the supine to the standing position and high-pressure peaks of more than 100 mm Hg during walking in some way mimicking the pressure amplitudes of an IPC device.

Our study seems to show that compression therapy can be safely applied in VLU and MLU also when non-insulin dependent diabetes mellitus coexists as it can deliver the described compression effects. Venous and mixed leg ulcers in diabetic patients heal when treated by compression devices even if diabetes seems to slow the healing process significantly in VLU patients, not significantly in MLU patients. Sex seems to play a role in MLU, (females showed a longer healing time compared with males) but even this figure must be confirmed by larger studies. All other considered variables (age, smoking habit, arterial hypertension, BMI) do not seem to play any role.

Limitations of this study: one limitation is the fact that this is a retrospective study, in which a priori sample size calculation as well as gender-balance cannot be done. This could affect the applicability of our findings. Another limitation is that we are not actually reporting data on compression therapy in diabetic foot or in diabetic ulcers, but in VLU or MLU with a coexistent diabetes mellitus. Despite these limitations we believe that our data support the indication to compression therapy in patients with VLU and MLU and co-existent diabetes.

## 5. Conclusions

This retrospective study demonstrates that no major differences should be expected concerning recalcitrant ulcer healing treated by CT between DP and NDP. All patients were treated by the same methods: inelastic bandages and sclerotherapy of superficial venous reflux. Our results indicate that CT may be safely applied in diabetic patients with recalcitrant ulcers, even in the presence of moderate PAOD. In addition, they show that compression in DP did not result in unwanted effects, but just a light, not significant healing delay compared to NDP even if these data need to be confirmed by prospective randomized control studies.

Basically, diabetes does not represent a contraindication to compression therapy in patients with VLU even with additional arterial occlusive disease (excluding critical limb ischemia) and does not play a negative prognostic role as to ulcer healing.

## Figures and Tables

**Figure 1 jcm-09-03709-f001:**
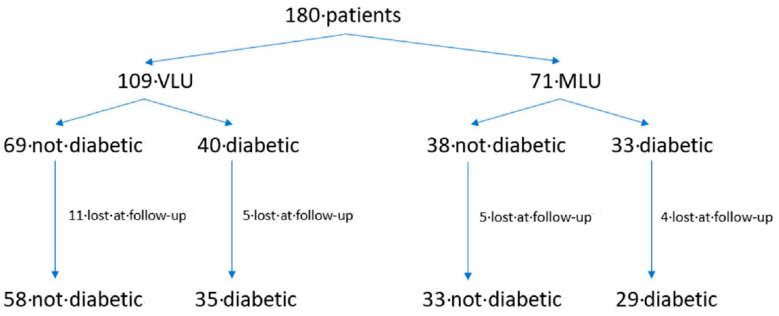
Consort Diagram of the case series.

**Figure 2 jcm-09-03709-f002:**
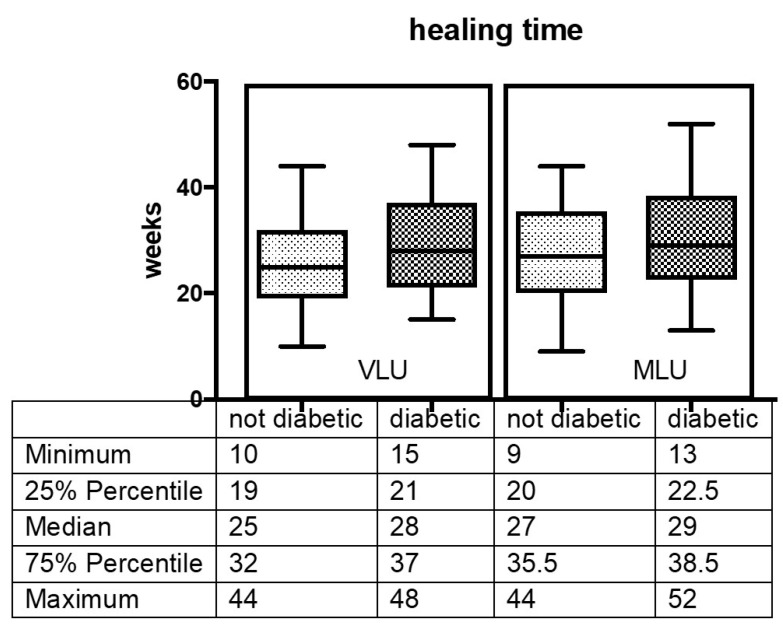
Ulcer healing time in diabetic patients and not diabetic patients with Venous Leg Ulcer and with Mixed Leg Ulcer.

**Figure 3 jcm-09-03709-f003:**
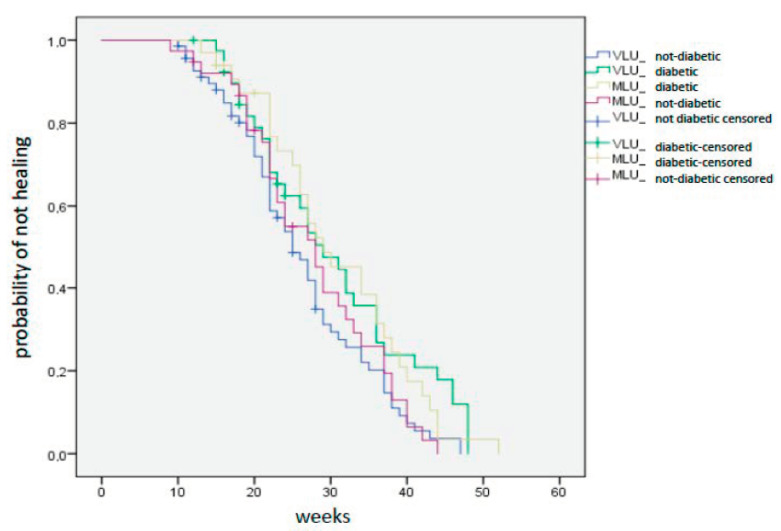
Venous Leg Ulcers and Mixed Leg Ulcers survival curves in diabetic and not diabetic patients.

**Figure 4 jcm-09-03709-f004:**
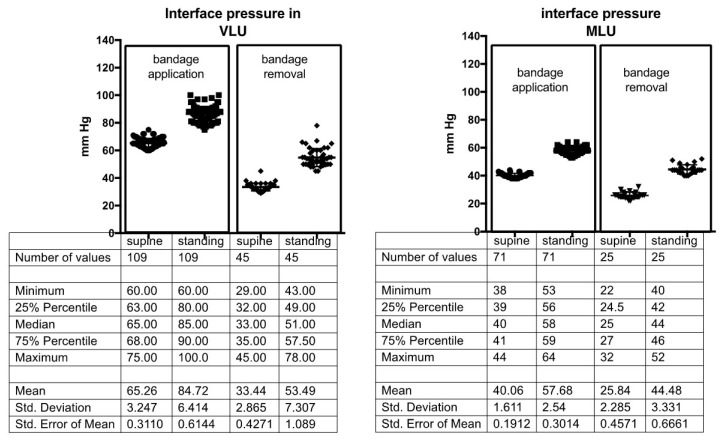
Compression pressure values in supine and standing position both at application and at removal of the bandages in patients with Venous Leg Ulcers and with Mixed Leg Ulcer.

**Table 1 jcm-09-03709-t001:** Case series demographic features.

	VLU		MLU	
	Not Diabetic	Diabetic	Not Diabetic	Diabetic
sex Males/Females	11/47	9/26	5/28	12/17
mean age (standard deviation)	73 (12.6)	71 (15.7)	77 (9.3)	74 (10.9)
Superficial Venous Insufficiency	36	23	21	16
Deep Venous Insufficiency	9	3	8	8
Superficial + Deep Venous Insufficiency	13	9	4	5
median ABPI (InterQuartile Range, IQR)	1 (1–1.1)	1 (1–1.1)	0.68 (0.61–0.73)	0.65 (0.6–0.72)
median ulcer surface cm^2^ (IQR)	40 (3–65)	40 (23–65)	39 (24–55)	40 (15.5–65)
median ulcer duration in months (IQR)	8 (7–18)	8 (6–18)	8 (6–19)	8 (6–16)
smoking habit	13	7	8	6
arterial hypertension	34	25	23	22
BMI > 30	14	15	8	12

VLU: venous leg ulcers. MLU: mixed leg ulcers.

**Table 2 jcm-09-03709-t002:** Multivariate Cox model analysis.

	Group							
	VLU				MLU			
	HR	95.0% CI HR		*p*	HR	95.0% CI HR		*p*
Sex Male vs. female	1.031	0.594	1.792	0.913	0.457	0.225	0.930	0.031
Age	0.996	0.979	1.013	0.625	1.001	0.970	1.033	0.962
Smoking habit	1.021	0.607	1.716	0.938	1.956	0.987	3.879	0.055
Arterial hypertension	0.822	0.514	1.313	0.412	0.678	0.355	1.292	0.237
BMI > 30	1.044	0.630	1.732	0.866	0.727	0.399	1.322	0.296
Ulcer surface	0.946	0.933	0.959	0.000	0.897	0.873	0.923	0.000
Not diabetic vs. diabetic	0.460	0.284	0.746	0.002	0.760	0.418	1.382	0.368

HR: Hazard Ratio, CI: Confidence Interval.

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
