# Peer review of "Compression Therapy Is Not Contraindicated in Diabetic Patients with Venous or Mixed Leg Ulcer"

_jcm, 2020, doi:10.3390/jcm9113709_

Round 1
Reviewer 1 Report
This is a very interesting study that has the potential to contribute to the knowledge of the field and advance treatment pathways.
However, I am surprised that you have missed two publications that delve in the VLUs pathophysiology and should be cited, as they would help your readers understand why CT works (i.e., Klonizakis M. The Role of Microcirculatory Dysfunction in the Pathophysiology and Treatment of Venous Leg Ulcers. JAMA Dermatol. 2019 Jul 1;155(7):861-862.doi: 10.1001/jamadermatol.2019.0591. This is not the only place that this happens; in the discussion you do the same when you mention "many trials" for which IDDM was an exclusion factor, but you only have one reference, which is not an RCT, but a review article. You also state that "Diabetes and related disorders, such as metabolic syndrome, are
directly linked with degeneration of arterial wall and the development of a general cardio-cerebro vascular risk" but you don't refer this to anywhere. In general it is best not to overcite, but I would go and look for recent and at least a couple of references, every time I make strong statements, which I would recommend that you do as well.
The other matter that is not clear is how you ended up with this number of participants: there isn't anywhere a clear and transparent statistical calculation. This combined with the absence of the effect sizes (which should be added), prevents as deduct how (and if) your findings are defining or indicative. If you haven't done a sample size, then you should report this as a major limitation.
A similarly important limitation is the fact that your groups are not gender-balanced. This is important due to the differences in prevalence and potentially pathophysiology of the condition. In fact I can't find (if this exists somewhere - I would ask to be forgiven) if there are any factors for which, the groups were balanced.
Also, although I am not in favour of lengthy discussions, I feel that you have cut short at many points. For example, the opening statement of the discussion is strong, but it is lost - for example why it is important to include patients in these trials? (We know but it needs to be spelt out, supported by numbers). I would advise that you try and revisit the discussion, expand where possible and explain more clearly the rationale and the importance of your finding and also suggest the "what's next".
Furthermore, how and where you recruited your participants, and where this was ethically approved, remains a mystery - please clarify.
There are also numerous grammar and punctuation mistakes - please read these again and be careful about using , and . in p values.
Author Response
This is a very interesting study that has the potential to to the knowledge of the field and advance treatment pathways.
Authors: thank you very much.
However, I am surprised that you have missed two publications that delve in the VLUs pathophysiology and should be cited, as they would help your readers understand why CT works (i.e. Klonizakis M. The Role of Microcirculatory Dysfunction in the Pathophysiology and Treatment of Venous Leg Ulcers. JAMA Dermatol. 2019 Jul 1;155(7):861-862.doi:10.1001/jamadermatol.2019.0591.
Authors: pathophysiology of VLU and how and why CT works is beyond the aim of this manuscript. In addition, it is so a huge topic that it would need an entire manuscript. We are confident that the readers of this highly respected journal know the pathophysiology of VLUs and why CT works in their management. Finally you mention just one paper (not two) and this paper, if I’m not wrong, is a letter to editor without any explanation on VLU pathophysiology or CT mode of action.
This is not the only place that this happens; in the discussion you do the same when you mention "many trials" for which IDDM was an exclusion factor, but you only have one reference, which is not an RCT, but a review article.
Authors: you are completely wright. We changed the reference. We enclosed in this revised version a Cochrane review concerning compression in VLUs. In this review diabetes was an exclusion criterium for compression in 18 out of 38 papers.
You also state that "Diabetes and related disorders, such as metabolic syndrome, are directly linked with degeneration of arterial wall and the development of a general cardio-cerebro vascular risk" but you don't refer this to anywhere. In general it is best not to overcite, but I would go and look for recent and at least a couple ofr eferences, every time I make strong statements, which I would recommend that you do as well.
Authors: thank you for your suggestion. Reference 7 was added.
The other matter that is not clear is how you ended up with this number of participants: there isn't anywhere a clear and transparent statistical calculation. This combined with the absence of the effect sizes (which should be added), prevents as deduct how (and if) your findings are defining or indicative. If you haven't done a sample size, then you should report this as a major limitation.
A similarly important limitation is the fact that your groups are not gender-balanced. This is important due to the differences in prevalence and potentially pathophysiology of the condition. In fact I can't find (if this exists somewhere - I would ask to be forgiven) if there are any factors for which, the groups were balanced.
Authors: we tried to better specify that in this manuscript we performed a sub-analysis of one of our previous retrospective study performed reviewing the folder of 180 consecutive patients treated in an outpatients setting from January 2011 to July 2014. This is the reason why there is no sample size calculation and the study is not gender-balanced. In addition we included the absence of sample size as major limitation of this study and wrote that data are “indicative” and must be confirmed by future larger RCTs.
Also, although I am not in favour of lengthy discussions, I feel that you have cut short at many points. For example, the opening statement of the discussion is strong, but it is lost – for example why it is important to include patients in these trials?(We know but it needs to be spelt out, supported by numbers). I would advise that you try and revisit the discussion, expand where possible and explain more clearly the rationale and the importance of your finding and also suggest the "what's next".
Authors: discussion was expanded and we think to have better highlighted the role of compression in VLU and MLU patients with coexistent diabetes
Furthermore, how and where you recruited your participants, and where this was ethically approved, remains a mystery – please clarify.
Authors: we hope that now the participant recruitment is more clear. Ethics statement was added
There are also numerous grammar and punctuation mistakes -please read these again and be careful about using , and . in p values.
Authors: this was done
Reviewer 2 Report
Thank you for the opportunity to review this paper - it addresses an important question and reassures this triallist that recruiting diabetic participants into our trials is the right thing to do.
I have a few minor suggestions, which are mainly stylistic.
- TITLE: add "venous or mixed" so that the title reads "Compression therapy is not contraindicated in diabetic patients with venous or mixed leg ulcers."
- Lines 23-25: The font is different.
- Lines 24-25: The presentation of p values is inconsistent. I also suggest only two decimal places are required rather than 4.
- Line 26: Use regimen rather than "regime".
- Lines 49-50: The font is different.
- Line 51: Suggest moving "co-existence" so the phrase reads "...account the co-existence of diabetes mellitus type II as a potential confounding variable."
- Line 97: UGFS is not defined prior to using the abbreviation.
- Line 100: Missing words, presumably "Ulcer pain..."
- Line 105: Add in leading zero to P value <0.05
- Line 113: 13 males and 17 females?
- Table II: Use full term rather than SVI, DVI, and SVI+DVI; Presumably the mean value has been reported for ABPI if st.dev is also reported, not the median value. Use actual values for ABPI rather than >1 (and report SDs for these values).
- Lines123-124: The presentation of p values is inconsistent. I also suggest only two decimal places are required rather than 4.
- Figure 2: Remove chart title, use English for the x-axis legend, and use "Probability of not healing" as the legend for y-axis.
- Line 157: Remove sensation.
- Line 167: Reword so that sentence reads "As a consequence, the co-existence of diabetes in VLU and MLU raise questions in terms of treatment and the current literature does not support any definitive approach."
- Line 169-170: Different font colour.
- Lines 184-186: I do not understand the need for this statement. Perhaps consider removing it?
- Line 188: Substitute "only" for "overall"
- Line 194: Use recent instead of new.
- Line 195: Remove "old" and "even" as they are redundant terms.
- Line 214-217: This paragraph does not seem to be necessary.
- I don't think this limitation is accurate as the study did not set out to examine the role of compression in DFU. I think the limitation of the study is that it is an observation design subject to the potential for confounding and the analysis not attempted to adjust for confoundings such as between-group difference in age, smoking, hypertension, and BMI. That said, I do not think that this limitation is one that should preclude publication.
- Line 221: Remove the word "scientific" as it is an unncessary descriptor for the study..
Author Response
Thank you for the opportunity to review this paper - it addresses an important question and reassures this triallist that recruiting diabetic participants into our trials is the right thing to do.
Authors: thank you
I have a few minor suggestions, which are mainly stylistic.
TITLE: add "venous or mixed" so that the title reads "Compression therapy is not contraindicated in diabetic patients with venous or mixed leg ulcers."
Authors: done
Lines 23-25: The font is different.
Authors: changed even if the version you have comes from the journal
Lines 24-25: The presentation of p values is inconsistent. I also suggest only two decimal places are required rather than 4.
Authors: amended
Line 26: Use regimen rather than "regime".
Authors: done
Lines 49-50: The font is different.
Authors: see above
Line 51: Suggest moving "co-existence" so the phrase reads "...account the co-existence of diabetes mellitus type II as a potential confounding variable."
Authors: done
Line 97: UGFS is not defined prior to using the abbreviation.
Authors: amended
Line 100: Missing words, presumably "Ulcer pain..."
Authors: thank you, you are right. The words were added
Line 105: Add in leading zero to P value <0.05
Authors: done
Line 113: 13 males and 17 females?
Authors: amended
Table II: Use full term rather than SVI, DVI, and SVI+DVI; Presumably the mean value has been reported for ABPI if st.dev is also reported, not the median value. Use actual values for ABPI rather than >1 (and report SDs for these values).
Authors: done
Lines123-124: The presentation of p values is inconsistent. I also suggest only two decimal places are required rather than 4.
Authors: done
Figure 2: Remove chart title, use English for the x-axis legend, and use "Probability of not healing" as the legend for y-axis.
Authors: done
Line 157: Remove sensation.
Authors: done
Line 167: Reword so that sentence reads "As a consequence, the co-existence of diabetes in VLU and MLU raise questions in terms of treatment and the current literature does not support any definitive approach."
Authors: done
Line 169-170: Different font colour.
Authors: see above. Not our fault
Lines 184-186: I do not understand the need for this statement. Perhaps consider removing it?
Authors: we think this statement is important also considering the changed reference.
Line 188: Substitute "only" for "overall"
Authors: done
Line 194: Use recent instead of new.
Authors: done
Line 195: Remove "old" and "even" as they are redundant terms.
Authors: done
Line 214-217: This paragraph does not seem to be necessary.
Authors: we think this paragraph is important because compression therapy counteracts the negative influence of diabetes in inflammatory mediators and this may contribute to the beneficial effects of compression. This was better specified in the text
I don't think this limitation is accurate as the study did not set out to examine the role of compression in DFU. I think the limitation of the study is that it is an observation design subject to the potential for confounding and the analysis not attempted to adjust for confoundings such as between-group difference in age, smoking, hypertension, and BMI. That said, I do not think that this limitation is one that should preclude publication.
Authors: we added a Cox analysis of the variables that shows that the variable as sex, BMI and so on do not seem to play any role.
Line 221: Remove the word "scientific" as it is an unncessary descriptor for the study.
Authors: done.
Round 2
Reviewer 1 Report
The authors have not addressed satisfactorily all of my comments. This needs to be rectified. More specifically,
1) The response that "the pathophysiology is a huge topic" isn't an satisfactory answer. You also cannot assume that the readers will be knowledgeable. A short paragraph is needed, because otherwise you cannot substantiate why compression works and what else would be required to assist the treatment process.
2) Limitations: You note that "One limitation is the fact that this is a retrospective study in which sample size calculations would be inadequate". This is not correct - it should be "this is a retrospective study in with no a priori sample size calculation". You should also note that this may affect the applicability of your findings. You should also move the fact that "these data need to be confirmed by prospective randomized control studies" in conclusions and change "we are convinced" to "we believe". The former isn't scientific.
You should also write specifically in limitations that the study isn't gender balanced.
3) Additionally, in the light of absence of a study-specific sample size calculation, I would request that effect sizes are calculated and added in the manuscript.
Author Response
The authors have not addressed satisfactorily all of my comments. This needs to be rectified. More specifically
The response that "the pathophysiology is a huge topic" isn't an satisfactory answer. You also cannot assume that the readers will be knowledgeable. A short paragraph is needed, because otherwise you cannot substantiate why compression works and what else would be required to assist the treatment process.
Authors: we are very sorry that you were not satisfied. The main objection of our previous response is that this explanation is beyond the scope of this work. To make it very simple if an engineer explains how to build a house nobody will ask him the time tables. Anyway, we tried to greatly summarize a topic that would request an entire long paper. We cannot guess if this short paragraph can be understood by readers who do not know what compression therapy is, its action modality, indications and contraindications, limits. As a consequence, we added some references so that, who is interested, can go more in depth on this topic.
Limitations: You note that "One limitation is the fact that this is a retrospective study in which sample size calculations would be inadequate". This is not correct - it should be "this is a retrospective study in with no a priori sample size calculation".
Authors: you are right, we did the change.
You should also note that this may affect the applicability of your findings. You should also move the fact that "these data need to be confirmed by prospective randomized control studies" in conclusions and change "we are convinced" to "we believe". The former isn't scientific.
Authors: thank you for your suggestions, we did both changes.
You should also write specifically in limitations that the study isn't gender balanced.
Authors: done
3) Additionally, in the light of absence of a study-specific sample size calculation, I would request that effect sizes are calculated and added in the manuscript.
Authors: done, added in methods